# Role of Iron Metabolism-Related Genes in Prenatal Development: Insights from Mouse Transgenic Models

**DOI:** 10.3390/genes12091382

**Published:** 2021-09-02

**Authors:** Zuzanna Kopeć, Rafał R. Starzyński, Aneta Jończy, Rafał Mazgaj, Paweł Lipiński

**Affiliations:** Institute of Genetics and Animal Biotechnology, Polish Academy of Sciences, 05-552 Jastrzębiec, Poland; z.kopec@igbzpan.pl (Z.K.); r.starzynski@igbzpan.pl (R.R.S.); a.jonczy@igbzpan.pl (A.J.); r.mazgaj@igbzpan.pl (R.M.)

**Keywords:** blastocyst, development, embryo, fetus, gene deletion, implantation, iron, mouse, prenatal

## Abstract

Iron is an essential nutrient during all stages of mammalian development. Studies carried out over the last 20 years have provided important insights into cellular and systemic iron metabolism in adult organisms and led to the deciphering of many molecular details of its regulation. However, our knowledge of iron handling in prenatal development has remained remarkably under-appreciated, even though it is critical for the health of both the embryo/fetus and its mother, and has a far-reaching impact in postnatal life. Prenatal development requires a continuous, albeit quantitatively matched with the stage of development, supply of iron to support rapid cell division during embryogenesis in order to meet iron needs for erythropoiesis and to build up hepatic iron stores, (which are the major source of this microelement for the neonate). Here, we provide a concise overview of current knowledge of the role of iron metabolism-related genes in the maintenance of iron homeostasis in pre- and post-implantation development based on studies on transgenic (mainly knock-out) mouse models. Most studies on mice with globally deleted genes do not conclude whether underlying in utero iron disorders or lethality is due to defective placental iron transport or iron misregulation in the embryo/fetus proper (or due to both). Therefore, there is a need of animal models with tissue specific targeted deletion of genes to advance the understanding of prenatal iron metabolism.

## 1. Introduction

Iron is an essential redox element that functions as a cofactor of many hemo- and iron-sulfur [Fe-S] proteins required to sustain fundamental life processes including DNA synthesis and repair, ATP production, cell cycle, oxygen transport, and detoxification. There is a widespread consensus that an accurate iron delivery to the sites of intracellular iron-sulphur ([Fe-S]) clusters biogenesis and heme molecules synthesis within the cells is a key distinguishing feature of intracellular iron homeostasis [1,2]. However, considering that the redox activity of iron can generate, through the Fenton reaction, hydroxyl free radicals capable of causing a wide range of biological damage [3], the second challenge of iron homeostasis is minimizing its toxicity. This dual homeostatic assignment is achieved via the functioning of a complex protein machinery involving transmembrane iron transporters, chaperones, iron storage proteins, ferrireductases, and ferroxidases. Cellular iron uptake, storage, efflux, and utilization are to a large extent coordinated and post-transcriptionally regulated via iron regulatory proteins (IRP1 and IRP2), the cytosolic iron sensors that bind iron-responsive elements (IRE) in regulated messenger RNAs [1]. The management of iron at the systemic level must adjust exogenous (dietary iron absorption by duodenal enterocytes) and endogenous (iron recycled from senescent red blood cells by macrophages) iron supply to satisfy the iron needs of the organism related mostly to the haemoglobinization of new red blood cells during the process of erythropoiesis. As there is no homeostatic mechanism for the excretion of excess iron from the organism, it is therefore clear that the control of total body iron content occurs at the level of dietary absorption. This meticulous regulation largely relies on the hepcidin and ferroportin interaction [4] but also involves hypoxia inducible factor-2, which regulates the expression of key genes that contribute to iron absorption [5]. The liver-derived hormone hepcidin, considered the central regulatory molecule of systemic iron homeostasis [6], controls systemic iron levels by blocking the iron-exporting protein ferroportin in duodenal enterocytes, hepatocytes and reticuloendothelial macrophages, the sites of iron absorption, storage and recycling, respectively [7,8]. By inhibiting major flows of iron into the circulation, hepcidin maintains an adequate level of iron bound to transferrin, (blood plasma protein, a major vehicle for iron delivery to cells) which is a hallmark of systemic iron balance.

This clipped outline of mammalian iron homeostasis is derived mainly from studies of patients with inherited iron homeostasis disorders and mutant animal models carried out over the last 20 years, depicting iron handling in adult organisms. Much less is known about how iron metabolism is balanced at various stages of prenatal development. Considering that the mammalian embryo undergoes a variety of changes as development proceeds from the one-cell to the pre- and post-implantation stages, and then to the fetus, the key focus of prenatal iron metabolism is quantitatively differentiated iron acquisition, which means that it depends on the developmental stage. The mammalian embryo/fetus grows extremely fast, and the embryos must produce more and more red blood cells to accommodate this growth. The rate of erythropoiesis in utero, depending on the stage of development, is a central determinant of iron needs. One of the main iron-metabolism pathways in prenatal life that has yet to be fully explored is unidirectional iron delivery to the embryo/fetus. It is believed that iron is not transferred in the reverse direction [9]. In contrast to well-defined pathways of intestinal absorption of exogenous iron operating postnatally, in prenatal life, iron reaches to the organism through the transfer across at first the visceral endoderm (VE), then structures called “early placenta” [8] and fully developed/specialized placenta. This process is the subject of a complex, regulatory interplay among the mother, fetus, and the placenta itself (reviewed in [10]).

The targeted disruption of iron metabolism genes in mice resulting in the retardation of embryonic development or causing embryonic lethality has proved to be a valuable tool for deciphering prenatal iron regulatory processes. This review compiles current data from transgenic and spontaneous mutant mouse models (Table 1), which have proven successful at providing insights into the molecular mechanisms of the maintenance of iron homeostasis in the embryo proper and placental iron transport. Considering that mouse models replicate the main features of human iron metabolism [11], and taking into account some similarities between murine and human placental development and structure [12], our review may provide insight into prenatal iron handling in humans.

## 2. Iron Handling in a Mouse Embryo during Preimplantation Development

The timeline of embryonic development is defined by the flow of time from the fertilization. Thus, in mice, the age of the embryo is usually expressed in half-day intervals since the time of mating, which is determined by the detection of a vaginal plug. Assuming that fertilization usually takes place around midnight and that checking for the plug is performed early in the morning, 0.5 post-coitum (post-coital age) or embryonic (E) day 0.5 (embryonic age) is considered to correspond with noon on the day on which the vaginal plug is found. In mice, preimplantation development of the embryo includes several stages from the fertilization of the oocyte (*ovum*) in the oviduct to the formation of the zygote, which then undergoes successively cleavage divisions to generate a blastocyst initially consisting of the trophectoderm and the inner cell mass, which subsequently differentiates into the epiblast and the primitive endoderm of yolk sac tissue and the embryo proper and the most of fetal membranes [27]. In the mice, expanded blastocyst of around one hundred cells implants in the uterus at E4.5 [28,29,30].

Although iron is required in high amounts for numerous critical bioreactions and biosynthetic pathways in rapidly dividing cells, including heme synthesis, respiratory chain function, and DNA synthesis, it seems that the iron demand of the embryo during the preimplantation period is well-balanced using endogenous sources. This may be due to high iron stores contained in the oocyte, which are sufficient to sustain early embryonic development. In the oocyte of xenopus laevis, iron does not vary throughout the first 50 stages of development, when all tadpole organs are forming. Hence, the complement of iron needed for incorporation into apoproteins during development is already present at a time when oocyte maturation is completed. It can therefore be considered that there is a sufficiently supply of iron in the oocyte [31]. In porcine embryo studies, iron deficiency leads to severe apoptosis in blastocysts, which may be explained by mitochondrial dysfunction [32]. Indeed, oocyte development takes place in an iron-rich environment provided by ovarian follicles [33]. Transferrin (Tf) has been identified as an important protein constituent of follicular fluid [33]. Furthermore, the expression of Tf and transferrin receptor 1 (TfR1) genes has been detected in the human granulosa cells that surround the oocyte within the follicle and function to produce sex steroids, as well as numerous growth factors thought to influence oocyte development [34]. The most abundant intracellular source of iron is ferritin (Ft), a ubiquitous cytoplasmic protein composed of a protein shell that can accommodate up to 4500 atoms of iron. It has the dual functions of iron detoxification and iron reserve. Ferritin is a heteropolymer composed of 24 subunits of two types: L (FTL1 for light) and H (FTH1 for heavy), depending on their relative molecular weights. The two subunit types have different functional properties. The H chain has an iron-binding site and exhibits ferroxidase activity, whereas the L chain, has a nucleation site involved in iron-core formation [35]. Iron released from Ft may involve ferritinophagy, a ferritin autophagic degradation process mediated by the selective nuclear receptor coactivator-4 (NCOA4). NCOA4 binds to ferritin and delivers it to nascent autophagosomes, which then merge together with the lysosomes causing Ft degradation and the release of iron that can be reused in cellular processes [36]. The result of the targeted deletion of the gene encoding H-Ft highlights its unique ferroxidase activity, which is indispensable in early embryonic development. The timeline analysis of the lethality of embryos homozygous for the non-functional *Fth1* allele (*Fth1*^−/−^) clearly indicated that they die between days E3.5 and E9.5 days of development. Genotyping of the E3.5 embryos (obtained from *Fth1^+/−^* females crossed with *Fth1^−/−^* males) revealed that *Fth1**^−/−^* blastocysts were present at the expected Mendelian frequency (~25%) and displayed a normal morphology [13]. These results strongly suggested that the autonomous growth of *Fth1^−/−^* embryos from fertilization to day E3.5 may rely on the use of maternal ferritin-derived iron present in the oocyte. On the other hand, this study show that the absence of a functional intracellular cytoplasmic ferritin is lethal to embryonic cells just after E3.5. The possible reason for this is that iron increasingly entering into the embryo cells after blastocyst implantation cannot be sequestered and detoxified, initiating the Fenton reaction and overproduction of highly toxic hydroxyl radicals. In contrast to the total early embryonic lethality of the *Fth1* gene deletion in mice, truncation of the *Ftl* gene is only partially (in about 50%) lethal and occurs at later stages of embryonic development, i.e., at E11.5–E13.5. Interestingly, surviving *Ftl^−/−^* embryos were phenotypically normal and survived until birth [14]. Although iron status in survived *Ftl^−/−^* embryos has not been examined, the results of this study indicate that expression of the H subunit can rescue the loss of the L subunit and that FTH1 homopolymers still have some capacity to sequester iron as previously demonstrated in in vitro studies [37].

Although the membrane transferrin receptor 1 (TfR1)-mediated endocytosis of the complex of transferrin bound iron is the major route of cellular iron uptake for most mammalian cells (in particular to erythroid cells) [38], it seems that the iron-delivery pathway is not of major importance for the progression of the development from the zygote to the blastocyst stage. Embryos with global deletion of the the *TfR1* gene die much later after implantation, i.e., by E12.5 [15]. However, the data do not allow us to determine whether embryos death is due to insufficient placental iron transport via embryonic TfR1 and maternal transferrin, impaired erythroid iron uptake by embryonic proteins, or both. The synthesis of embryonic transferrin starts by the E7.5 day of gestation mainly in the yolk sac and liver [39]. Interestingly, *hpx* mice with splicing defects in the *Tf* genes that fail to express appropriate transferrin levels (producing less than 1% of the normal level of serum transferrin) are viable at birth, though they are profoundly anemic (they develop severe microcytic hypochromic anaemia) and can only survive for up to two weeks after birth [40]. It is not excluded that in mice lacking transferrin, non-transferrin bound iron (which usually appears in the blood plasma when transferrin becomes fully saturated with iron [41], could be a likely iron source for the fetus in these conditions. These findings provide strong evidence that the transferrin-iron utilization through the TfR1-transferrin pathway is indispensable for iron delivery to embryonic/fetal erythroid cells, and that transferrin is the major physiological iron source for prenatal erythropoiesis.

Lactoferrin (Lf), a glycoprotein of the transferrin family found in mucosal secretions, possesses a wide range of functions including iron-binding and transferring functions [42]. In mice, uterine expression of lactoferrin (LF) during the preimplantation period has been demonstrated from days one to eight of pregnancy [43]. Mammalian Lf receptors (LfR) play pivotal roles in mediating multiple Lf functions [44]. Although the Lf-LfR pathway has been suggested to deliver iron to the pre-implantation embryo [18], the presence of LfR at this stage of prenatal development has not been examined yet. LfR may be important for the function of Lf in the late post-implantation stages [45].

The iron regulatory proteins (IRP1 and IRP2) are two cytoplasmic RNA-binding proteins involved in the mechanisms that control iron metabolism in mammalian cells. They modulate the expression of iron-related proteins at a post-transcriptional level by binding to specific iron regulatory elements (IREs) on their mRNAs. IRP-IRE interaction can block protein synthesis or stabilize the mRNA. At low intracellular iron concentration, IRPs bind to the IRE of the ferritin subunits or ferroportin mRNAs and block their translation. Direct interactions between IRPs and IRE motifs stabilize TfR1 and DMT1 (divalent metal transporter 1) mRNAs. The converse regulation of Ft subunits and TfR1 synthesis, being a consequence of the lack of binding of IRPs to IRE, occurs in cells with high iron levels. Thus, IRP-mediated regulation rapidly restores the physiological levels of iron both when it is deficient and in excess. Animal models of IRP deficiency have proven useful for defining the specific as well as redundant functions of IRP1 and IRP2 under basal conditions [46,47] or in response to iron fluctuations [48]. Neither deletion of the *Irp1* nor the *Irp2* gene is embryonically lethal although mice with targeted deletion of IRP2 display microcytic anemia and iron mismanagement in most tissues. This shows us that IRP2 is critical for maintaining the iron balance in vivo [46,47]. In contrast, IRP1-null mice exhibit no spontaneous iron misregulation [47,48]. However, mice that completely lack both IRPs cannot survive through the gestation. The embryonic lethality of double knock-out manifests at the blastocyst stage before implantation [18]. A possible reason for the early death of *Irp1^−/−^*/*Irp2^−/−^* is functional iron deficiency. This means that in the absence of IRPs (repressors of H- and L-Ft translation) ferritin is abnormally up-regulated: thus, iron sequestered in the Ft shell may not be available for the metabolic needs of the blastocyst.

## 3. Iron Metabolism Proteins in Mouse Post-Implantation Development

The increasing nutrients (including iron) needs of the mammalian embryo during post-implantation development are met by the decidua and later placenta, a specialized organ that enables the exchange of nutrients, gases and wastes exchange between the maternal and embryonic/fetal compartments. The optimal transfer of iron across the placenta is essential for fetal development in utero and for the establishment of adequate birth iron stores to sustain growth in early infancy. However, in mice, the mature placenta appears at E14.5 [30]. Prior to placenta formation, the extraembryonic visceral endoderm functions in maternoembryonic nutrient (including iron) transport prior to placenta formation [49].

During mice implantation (E4.5), the mural trophectoderm surrounding the blastocyst cavity differentiates into nondividing primary trophoblast giant cells. The polar trophectoderm, adjacent to the inner cell mass, proliferates to form the ectoplacental cone and develops multitude of secondary giant cells. Post-implantation, the spherically shaped blastocyst transforms into the cylindrical. During this process, the polar TE generates the ExE (the extraembryonic ectoderm) [50]. After implantation, the primitive endoderm differentiates into the parietal and visceral endoderm. The former lies adjacent to giant trophoblast cells and gives rise to the endoderm of the parietal yolk sac. The second lineage, the visceral endoderm, which lies adjacent to the extraembryonic mesoderm gives rise to the visceral yolk sac, and takes part in primitive hematopoiesis and vasculogenesis [51,52]. In mice, the mature E14.5 placenta is composed of the maternal decidua on the outside, the junctional zone, and the innermost labyrinth [53]. In the decidua, the endothelium is replaced by trophoblast cells, and promotes the transition to hemochorial placenta [54]. The junctional zone at the implantation site consists of spongiotrophoblast cells and giant trophoblast cells. Nutrient exchange occurs in the labyrinth. It is composed of the trophoblasts of two layers of multi-nucleated syncytiotrophoblast [55].

The iron resources of the embryo are largely insufficient for sustaining its development during the post-implantation period. Therefore, the mechanisms of the continuous supply of extraembryonic iron from the maternal environment must be triggered at around day E4.5 to balance growing embryo iron requirements. Our current understanding of the molecular mechanisms by which iron is transferred across the mature, vascularized placenta has been the focus of recent reviews [10,56]. Briefly, iron metabolism genes that have roles in the maintenance of iron homeostasis in the adult often function in embryo proper and placental development as well. In the term placenta the unidirectional iron flux mediated by polarized syncytiotrophoblasts involves apical and basal transporters such as TfR1, DMT1, ZIP8, ZIP14 and ferroportin, respectively [10,56,57]. Much less is known about iron transport to the embryo/fetus at the earlier stages of placental development, from the initiation and maintenance of the trophoblast lineage to the complex functions of the mature placenta. The main insights into the molecular mechanisms of iron metabolism in the fetus have come from the study of various mouse models with disrupted iron metabolism genes. If a gene is indispensable for the embryo proper and placental iron transport, the targeted mutant embryo might die before later phenotypes become apparent. However, the global deletion of a gene resulting in the embryonic lethality does not allow us to conclude whether it is indispensable for placental iron transport or for the handling of iron in the embryo proper, or both. Therefore, selective inhibition of a gene in the embryo proper (as in the case of the *Slc40a1* gene coding for ferroportin [17]) or in the placenta is needed to precisely determine its role in the prenatal development. The delivery of iron to most cells occurs following the binding of transferrin-to-transferrin receptor 1 (TfR1) on the cell membrane. The transferrin-TfR1 complexes are then internalized by endocytosis, and iron is released from transferrin via a process involving endosomal acidification [58]. Released iron, before being exported out of endosome, is reduced from Fe^3+^ to Fe^2+^ by Steap3, a member of a unique family of Steap (six transmembrane epithelial antigen of the prostate), the transmembrane reductases [59]. The expression of TfR1 becomes evident in early post-implantation mouse embryos. At E6.5 its presence has been detected in the ectoplacental cone and mural trophoblast giant cell populations, as well as in the embryonic ectoderm. With development, TfR1 is progressively lost from trophoblast giant cells and at around E8.5, it is undetectable as these cells reach their terminal differentiation state. In contrast, the rapidly proliferating cells of the extra-embryonic ectoderm continue to express high levels of TfR1. In the E10.5 placenta, TfR1 is expressed primarily on the differentiated labyrinthine trophoblast cells involved in the maternal-fetal transfer of iron. Among mammalian cells the highest TfR1 level has been found in nucleated erythroid precursors and placental syncytiotrophoblasts [58]. Furthermore, TfR1 is highly up-regulated in the placenta under conditions of maternal iron scarcity [19,60]. All these data suggest that TfR1-mediated iron-transferrin uptake from the maternal source plays an important role in early embryonic post-implantation development and beyond. Embryos homozygous for a null allele of TfR1 (*TfR1^−/−^*) die by E12.5. However, some of them are not anemic and show normal number of circulating hemoglobinized erythroid cells in their vasculature as late as E10.5. implying that the transferrin-TfR1 pathway may not be essential for erythropoiesis during early development.

Divalent metal transporter (DMT1), encoded by the *Slc11a2* gene, is a divalent metal-ion transporter conserved from prokaryotes to higher eukaryotes that exhibits an unusually broad substrate range of divalent metals, including Fe^2+^, Zn^2+^, Mn^2+^, Cu^2+^, Cd^2+^, Co^2+^, Ni^2+^, and Pb^2+^, and mediates pH-dependent active proton-coupled transport [61]. DMT1 is the apical membrane iron transporter found in the epithelial cells of the duodenum, renal proximal and/or distal tubules. It is also the endosomal iron transporter in the transferrin cycle of developing red blood cells in the bone marrow and other cells [61]. It is also expressed in the placenta, where it potentially participates in materno-fetal iron transfer [57,62]. Microcytic anemia (*mk*) mice [63] and the Belgrade (*b*) rat [64], showing inherited defects in intestinal iron absorption and erythroid iron utilization that result in iron deficiency anemia, have the same spontaneous missense mutation (G185R) in *Slc11a2*. Similarly, mice with global deletion of the *Slc11a2* gene are born anemic, although iron deficiency anemia in DMT1 knock-out mice is more severe than that observed in animals homozygous for the G185R mutation [16]. Interestingly, at birth, total body iron content is normal, and hepatic iron stores are even increased in *Slc11a2^−/−^* mice compared to wild-type animals. This indicates that DMT1 is dispensable for efficient iron materno-fetal iron transfer. On the other hand, it is necessary for normal iron uptake by erythroid precursor cells, and thus, for the correct course of fetal erythropoiesis. Other potential apical transporters involved in iron trafficking are the ZRT/IRT-like protein (ZIP)14 (ZIP14; encoded by the Slc39a14 gene) and ZIP8 (encoded by the Slc39a8 gene). Both ZIP14 and ZIP8 are able to transport iron and expressed in mouse placenta [65]. ZIP14 also has been shown to mediate plasma membrane uptake of non-transferrin-bound iron as well [66]. Slc39a14^−/−^ do not different phenotypically from controls except for lower birth weight [67]. During mouse pregnancy, the level of placental Zip8 mRNA expression increases with development [67]. Deletion of the Slc39a8^−/−^ gene in mice leads to preweaning mortality [65]. This suggests more important role of ZIP8 in the iron homeostasis than ZIP14 and DMT1.In vertebrates, ferroportin (Fpn), the only known cellular iron exporter encoded by the *Slc40a1* gene, plays a pivotal role in cellular iron release, a function underlying systemic iron-metabolism processes, such as iron absorption by duodenal enterocytes, the recycling of iron derived from senescent erythrocytes by reticuloendothelial macrophages, and iron release from stores in hepatocytes [7]. Fpn has also been assigned the role of iron transporter in the placenta [17]. Indeed, Fpn is abundantly expressed along the basal membrane of human term placenta syncytiotrophoblasts [68] and mouse labyrinthine trophoblasts [69]. Furthermore, the strong up-regulation of placental Fpn expression during course of gestation (from day E12.5 to day E18.5) clearly indicates its importance in meeting the accelerating iron requirements of the developing embryo/fetus [70,71]. Importantly, studies in mice with a targeted disruption of the *Slc40a1* gene indicated a crucial and non-redundant role in iron supply to the developing embryo at the very early stage after implantation. The complete targeted disruption of the *Slc40a1* gene in mice, was lethal before E7.5, likely due to iron deficiency [17]. Immunohistochemical staining for ferroportin in a wild-type E7.0 embryo clearly indicated the basolateral expression of Fpn in the polarized epithelial cells of the extraembryonic visceral endoderm (exVE), while similar staining in a *Slc40a1^−/−^* showed no evidence of Fpn presence. Interestingly, disruption of the *Slc40a1* gene in all tissues except exVE rescued embryonic lethality, thus confirming the essential role of ferroportin in iron export to the embryo through this “early placenta” structure. Studies in transgenic animals with aberrant Fpn1 expression provide further evidence for the role of Fpn in maternal-fetal iron transport. Mouse embryos with a hypomorphic mutation in Fpn1 are severely iron deficient at E12.5 and exhibit defects in neural tube closure and forebrain patterning [69].

As already mentioned above, hepcidin, an iron-regulatory hormone produced mainly by the liver, controls plasma iron concentrations, tissue iron content and distribution by inhibiting major flows of this microelement into the circulation [4]. Transfer of iron across the placenta to the fetus may be regulated by maternal, placental and fetal hepcidin. The downregulation of maternal hepatic hepcidin expression and the lowering of its plasma level during pregnancy [10,70], increase iron bioavailability, and thus enhancing iron transport mediated by TfR1 and DMT1 localized on the apical side of the placental syncytiotrophoblasts, facing maternal circulation. On the other hand, fetal hepcidin would be expected to downregulate ferroportin on the basolateral side of these cells, facing fetal circulation. However, the predominant view is that fetal hepcidin does not significantly affect placental ferroportin in normal pregnancies simply because its concentration is very low [72]. Although fetal hepcidin mRNA expression has been show to increase between E13.5 and E17.5, nevertheless, at the end of pregnancy (E17.5–E18.5), it was much lower than that of maternal liver hepcidin [20,60]. A recent study, in which mouse fetuses with a liver-specific knockout of the *Hamp* gene were analysed, clearly indicates that placental ferroportin is not regulated by fetal hepcidin under normal physiological conditions [20,70]. It is possible that under certain pathological conditions, i.e., in inflammation (which is known to increase hepcidin expression [73]) the level of fetal liver hepcidin is high enough to induce placental ferroportin degradation and thus inhibiting placental iron transport [74]. Such a possibility arises from the a study carried out on transgenic mouse fetuses (E15.5) ubiquitously overexpressing the *Hamp* gene, which showed severe iron deficiency anemia [21].

A similar phenotype has been found in the type II transmembrane serine protease matriptase-2-null fetuses at E17.5 [22]. Matriptase-2 (encoded by the *Tmprss6* gene) is a repressor of hepcidin expression operating through the proteolytic degradation of membrane hemojuvelin [75,76], which in turn acts as a co-receptor and is required to fully activate transcription of the *Hamp* gene through the BMP/SMAD (bone morphogenetic protein/sons of mothers against decapentaplegic) signalling pathway [77]. Adult *Tmprss6^−/−^* mice were shown to excessively express the *Hamp* gene, leading to hypochromic, microcytic iron-deficiency anemia [75,78]). E17.5 *Tmprss6^−/−^* fetuses showed a concerted pattern of regulations such as strongly up-regulated liver hepcidin and decreased placental ferroportin expression, total non-heme body iron, and red blood cell indices [22]. This study highlights the fundamental role of constitutively expressed matriptase-2 in hepcidin suppression in fetuses to ensure iron mobilization, preventing iron deficiency and anemia.

## 4. Heme-Related Genes in Mouse Prenatal Development

Heme, a ferrous iron protoporphyin IX complex, is employed as a prosthetic group in diverse proteins that participate in important biological processes [79]. It is also involved in the transcriptional and translational regulation of the expression of numerous genes [80]. However, unbound heme is toxic due to its capacity to react with oxygen and catalyse the production of free radicals, which in turn may cause severe oxidative damage [81]. Therefore, the intracellular content of heme must be strictly regulated. One of the regulatory pathways is the enzymatic breakdown of heme molecules. In cells with an intensive heme metabolism such as erythroid cells and macrophages, Hmox-1 encodes for heme oxygenase 1 (HO1), a multifunctional, inducible enzyme, acting at the interface between heme and non-heme iron metabolisms catalyses the rate-limiting step in the heme degradation pathway by cleaving the porphyrin ring, at the expense of molecular oxygen. This releases ferrous iron, carbon monoxide (CO), and biliverdin [82]. Disturbances in iron metabolism at various stages of postnatal development are well characterized consequences of the *Hmox1* gene deletion. The effects of this deletion include impaired heme recycling by tissue macrophages, intravascular hemolysis, and severe anemia [83,84,85]). Undoubtedly, HO1 is also crucial for supporting prenatal development already from the early stages. In mice, HO1 expression has been found at the fetomaternal interface as early as the time of blastocyst implantation [23]. HO1 expression determines the attaching ability of blastocysts to endometrial epithelial cells and promotes the further formation of the placenta and its function. HO1 is markedly expressed in early embryos (HO-1 protein expression has been already shown in the ectoplacental cone in E6.5 embryos) and the placenta up to E14.5 and decreases towards the end of the pregnancy [23]. After placenta formation is completed on day E14.5, HO1 becomes restricted only to trophoblastic cells [86]. Murine invasive trophoblast is critically involved in placentation. It is not surprising therefore, that the deletion of the *Hmox1* gene leads to the abnormal placentation, inadequate remodelling of spiral arteries, intrauterine growth restriction, and eventually, to fetal lethality (reviewed in [87]). However, all these pathological consequences of HO1 deficiency seem unrelated to the misregulation of either cellular or systemic iron metabolism. Accordingly, the protective of HO1 effects on placentation and fetal growth can be mimicked by the exogenous administration of carbon monoxide (CO), a product of heme degradation catalysed by HO-1 [23].

It has been proposed that the placenta may utilize heme iron sources for fetal growth following the uptake of heme from the circulation [56]. Hemopexin is the protein produced mainly by the liver and released into plasma, where it binds heme with high affinity and acts primarily to deliver it to cells via CD91 receptor-mediated endocytosis [88]. During pregnancy, the high expression of CD91 in the placenta [88] may contribute substantially to the clearance of the heme-hemopexin complex from the circulation. The expression of HO1 in the placenta is well documented [87] and it is therefore likely that iron derived from the uptake of hemopexin-heme by CD91 is delivered to the fetus after heme degradation. However, the physiological relevance of this pathway is unclear and requires further study.

Another aspect of prenatal heme metabolism refers to the process of embryonic/fetal erythropoiesis. Erythropoiesis is the largest consumer of iron in the embryo/fetus during the prenatal period. It was estimated that in the mice fetuses between E12.5 and E16.5, the increase in red blood cell count is approximately 70-fold [89]. Erythropoiesis during development in utero is essential to supporting embryo survival, growth, and development of the embryo [90]. Murine intrauterine erythropoiesis consists of the primitive and definitive phases, both taking place in different sites of the embryo and fetus. Primitive erythropoiesis in the early fetal development, between E7.5 and E11.5, takes place entirely in the yolk sac blood islands. Definitive erythropoiesis begins during late fetal development, around E14.5 and occurs in the liver. Towards the end of gestation (E18.5), erythropoiesis passes from the liver to the bone marrow. Primitive and definitive erythropoiesis involve cell trafficking through erythroid progenitors, erythroblast precursors, and RBCs compartments.

The majority of heme (−85%) is synthesized in erythroid cells mainly for hemoglobin production. The heme biosynthetic pathway is composed of eight enzymes that work in either mitochondria or the cytoplasm. The terminal step of heme synthesis is the insertion of ferrous iron into the protoporphyrin IX macrocycle to produce protoheme IX (heme). This is catalysed by the enzyme ferrochelatase [91]. Growing evidence indicates that the tight control of cellular heme levels is not only achieved both by a fine balance between heme biosynthesis and catabolism via the enzyme heme oxygenase and also by specific transmembrane heme transporters [92]. Feline Leukemia Virus subgroup C Receptor (FLVCR) was identified as a mammalian cell surface heme exporter that appears to protect erythroid progenitors from potential heme toxicity during the heme synthesis phase of erythropoiesis [93]. The critical importance of the *Flvcr* gene for prenatal erythropoiesis and its consequence for embryo/fetal development has been experimentally proven in the study, in which interbred *Flvcr^+/−^* animals yielded no null offspring, attesting to the embryonic lethality of *Flvcr^−/−^* embryos/fetuses [24]. Intrauterine deaths occurred at two distinct periods of embryonic development: at or before E7.5 and between E14.5 and E16.5, corresponding to primitive and definitive erythropoiesis occurring in the yolk sac and in the liver, respectively [94]. The fetal death during the latter period is possibly due to deficient red cell production attributed due to the blocking at the proerythroblast stage (characterized by intensive heme synthesis), before hemoglobinization. It is hypothesized that FLVCR may protect proerythroblasts from heme toxicity, by expelling heme molecules into the extracellular environment [24].

## 5. Conclusions

There are still many more questions to answer about the regulation of iron homeostasis in prenatal life. In order to fully understand prenatal iron balance, it is crucial to distinguish between the mechanism and regulation of iron transport across the placenta, not only between maternal and fetal regulation but also between embryo proper vs. extraembryonic structures (such as embryonic part of the placenta and yolk sac). Although recent studies have revealed specific and unique molecules of iron transfer across the term placenta [10,56], it is clear that the supply of maternal iron to the embryo/fetus through immature placental structures at earlier stages of prenatal development should deserves our attention. Elucidating details of complex iron pathways during prenatal development can be greatly aided by mouse models with conditional knockout of genes in the placenta (or even in various placental cell types) but also mouse embryonic-extraembryonic chimaeras engineered to overexpress or to lack key iron metabolism genes in specific lineages and their derivatives. Similarly, many aspects of embryonic and fetal iron balance (for example, an adequate iron delivery for erythropoiesis occurring at different sites of the embryo/fetus) merit further investigations.

## Figures and Tables

**Table 1 genes-12-01382-t001:** Impact of the deletion of iron metabolism genes in mice on prenatal development.

Gene	Gene Product	Function	Phenotype	Reference
*Fth1*	Ferritin (Ft) H-subunit	Ferroxidase activity, essential for iron uptake by the ferritin molecule	Lethality from E3.5 to E9.5	[13]
*Ftl1*	Ferritin (Ft) L-subunit	Has a nucleation site involved in iron-core formation inside the protein envelope	Partial (about 50%) lethality at E11.5–E13.5	[14]
*Tfr1*	Transferrin receptor 1	Import of iron from transferrin into cells by endocytosis	Lethality after implantation, by E12.5 Affected both erythropoiesis and neurologic development	[15]
*Slc11a2*	Divalent metal ion transporter 1 (DMT1)	Transport of ferrous iron (Fe^2+^) and some divalent metal ions across the plasma membrane and/or out of the endosomal compartment.	No data about prenatal development. Newborn mice are anemic, without developmental abnormalities.	[16]
*Slc40a1*	Ferroportin (Fpn)	Transport of iron from the inside of a cell to the extracellular environment.	Lethality around E7.5 in embryos with global KO. Rescue of embryonic lethality through selective KO of ferroportin in the embryo proper	[17]
*Aco 1 (Irp1) Irp2*	Iron Regulatory Proteins 1 and 2	Role in post-transcriptional regulation of several mRNAs encoding iron metabolism proteins	No overt abnormalities in either Irp1 nor Irp2 knockout embryos/fetuses Lethlity of double Irp1 and Irp2 KO embryos at E6.5 and beyond. Functional iron deficiency	[18]
*Hamp1*	Hepcidin	Central regulator of systemic iron homeostasis. Role in the regulation of the entry of iron into the circulation	Global hepcidin knock-out has no effect on placental or fetal liver iron status in iron-replete or iron-deficient pregnancies. Reduction of hepatic iron content, decrease in hemoglobin concentration in liver-specific hepcidin KO embryos Severe microcytic anemia in fetuses with transgenic ubiquitous overexpression of the *Hamp* gene	[19,20,21] ^1^
** *Tmprss6* **	Matriptase-2	Suppressor of hepatic hepcidin expression	Strongly up-regulated liver hepcidin and decreased placental ferroportin expression. Reduction in total non-heme body iron, and some red blood cell indices in E17.5 fetuses, halmarking iron deficiency and microcytic anemia	[22]
*Hmox1*	Heme oxygenase 1 (HO1)	Role in in the enzymatic breakdown of heme molecules	Abnormal placentation, inadequate remodeling of spiral arteries, intrauterine growth restriction, and eventually fetal lethality	[23]
*Flvcr1*	Feline leukemia virus subgroup C receptor-related protein 1	Export of cytoplasmic heme to the outside of the cell	Deficient red cell production. Lethality at one of two embryonic times: at or before E7.5 and between E14.5 and E16.5.	[24]
*ISCA1*	Iron-sulfur cluster assembly 1	Iron-sulfur cluster (Fe-S) carrier, accepting (Fe-S) from a scaffold protein and transferring it to target proteins	Lethality at E8.5 and beyond Damage to the electron transport chain and TCA cycle.	[25] ^2^
*Fxn, Frda*	Frataxin	Precise function remains unclear. Involvement in Fe-S cluster and heme synthesis, energy conversion and oxidative phosphorylation, iron handling and response to oxidative damage	At E7.5 and E8.5 embryos start to be resorbed and reduced to a small mass of embryonic tissue surrounded by maternal hemmorrhagic tissue Complete resorption at E9.5.	[26]

^1^ Overexpression of the gene; ^2^ knockout of the gene performe in rats.

## Data Availability

Not applicable.

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
