# Peer review of "Role of Iron Metabolism-Related Genes in Prenatal Development: Insights from Mouse Transgenic Models"

_genes, 2021, doi:10.3390/genes12091382_

Round 1

Reviewer 1 Report

This is an interesting review of the metabolism of iron ions during pre- and postimplantation mammalian development. The authors focused on the detailed presentation of conclusions which can be drown form the studies in which phenotypes of knock-outs of the genes known to be involved in iron ions transport and metabolism were described, mostly in the mouse. This well written and comprehensive article may appeal to researchers interested in different aspects of iron biology in eukaryotes, and also will be of interest for mammalian developmental biologists. Accordingly, this text can be published in special issue of Genes dedicated to the different aspects of genetic regulation of iron homeostasis. Some minor improvements to the text are recommended:

l. 31 for the first time in text it is better to write “iron-sulphur ([Fe-S]) clusters” instead of “[Fe-S] clusters”;

l. 88 - 89 should be “endoderm of the yolk sac … and the embryo proper and the most of fetal membranes”;

l. 110 word “physiological”, which is not necessary, may be deleted;

l. 113 instead of “die during days E3.5 and E9.5 days of development” is better to put it as “die between days E3.5 and E9.5 of development”;

l. 157 please, expand and explain here the abbreviation DMT1, which appears for the first time in this line of text, but is explained much later, in the line 232;

l. 128 “fga” – what does it mean?;

l. 183 -184 there is no invagination of polar TE during the formation of the egg cylinder. Proliferating, solid extraembryonic ectoderm pushes down the epiblast, which becomes positioned on the tip of the early rod shaped egg cylinder covered by primitive proximal endoderm. Thus respective sentence should be rewritten.

Author Response

  1. 31 for the first time in text it is better to write “iron-sulphur ([Fe-S]) clusters” instead of “[Fe-S] clusters”;

Changed according to the Reviewer’s suggestion.

  1. 88 - 89 should be “endoderm of the yolk sac … and the embryo proper and the most of fetal membranes”;

Corrected according to the Reviewer’s suggestion.

  1. 110 word “physiological”, which is not necessary, may be deleted;

Deleted as suggested by the Reviewer.

  1. 113 instead of “die during days E3.5 and E9.5 days of development” is better to put it as “die between days E3.5 and E9.5 of development”;

Changed as suggested by the Reviewer.

  1. 157 please, expand and explain here the abbreviation DMT1, which appears for the first time in this line of text, but is explained much later, in the line 232;

DMT1 abbreviation has been expanded.

  1. 128 “fga” – what does it mean?;

„fga” has been deleted.

  1. 183 -184 there is no invagination of polar TE during the formation of the egg cylinder. Proliferating, solid extraembryonic ectoderm pushes down the epiblast, which becomes positioned on the tip of the early rod shaped egg cylinder covered by primitive proximal endoderm. Thus respective sentence should be rewritten.

Changed as suggested by the Reviewer.

Reviewer 2 Report

In this review, Kopec et al describe the current knowledge about the genes involved in iron homeostasis in mouse development. This review provides an interesting perspective of iron homeostatic genes in pre-implantation and post-implantation development. The authors rely largely on studies of transgenic (mainly knock-out) mouse models. The review is generally well written but contains several grammatical and spelling errors. Furthermore, the authors should include discussion on a broader selection of genes involved in iron homeostasis (see suggestions below) in order to provide a more comprehensive overview.

  1. Page 1, lines 30-32: Citation #1 (Wilkinson and Pantopoulos Blood 2013) is about IRP and HIFs. The authors should consider adding a different reference, such as a review, which would be more appropriate.
  2. Page 2 lines 49-50: Ferroportin is expressed on many other cell types that contribute to systemic iron balance in addition to the duodenum and macrophages. The authors should include the release of iron from liver stores as another major source of iron export to plasma.
  3. Page 2 lines 59-61: What do the authors mean by “quantitatively differentiated iron acquisition?” This sentence should be clarified. Do the authors mean that the iron requirements depend on the developmental stage? The next sentence should also be clarified: “erythropoiesis” itself means the production of red blood cells so it is redundant to say erythropoiesis must produce more and more red blood cells.
  4. Page 2 line 75: The authors should consider revising the statement that mice strongly replicate human iron metabolism. There are fundamental differences in iron metabolism between mice and humans. Humans rely primarily on iron recycling to meet daily requirements whereas mice depend mostly on dietary iron. However, the key regulatory and transport mechanisms are what is similar to humans.
  5. Page 3 first paragraph: The authors should use the official gene symbols for FTH1 (Fth1) and FTL (Ftl) and be consistent with the acronyms.
  6. Page 3 lines 93-96: Is it known that the embryo meets its own iron demands in the preimplantation period? The iron maternal requirements during the first trimester are lower than in menstruating women (Bothwell et al Am J Clin Nutr 2000), so is it that the mother has sufficient iron stores during the preimplantation period to meet early embryo iron demands? Similarly, the authors need to provide citations for the statements that the oocyte contains high iron levels, and that development takes place in an iron-rich environment.
  7. Page 3 lines 137-141: Although the Hpx null mice die shortly after birth, they are born viable which suggests another iron source is utilized for development when transferrin is not available (unless there is residual transferrin still being produced that is enough to support development). The authors should at least mention that non-transferrin bound iron could be a likely iron source for the fetus in these conditions, however the mechanisms of NTBI transport are not known. NTBI is detectable in the fetus at least in the first trimester (Evans et al Mol Hum Reprod. 2011 Apr; 17(4):227-32).
  8. Page 4 lines 177 and 189: The authors state that the mature placenta appears by E14.5 in mice, however the mouse placenta is fully formed by E10.5.
  9. Page 5 line 203: It is not definitively known that DMT1 is the endosomal iron transporter in placenta, it could still be ZIP8 or ZIP14.
  10. Page 5 line 216: Iron released from transferrin is the reduced before being exported out of the endosome. The candidate ferrireductases in the placenta should be mentioned (i.e., STEAP3, STEAP4)
  11. Page 5 last paragraph: In addition to DMT1, there are other potential transporters involved in iron trafficking (i.e., ZIP8, ZIP14) that are worth mentioning. Whereas ZIP14 appears to have a redundant role similar to DMT1, ZIP8 global knockout is embryonic lethal at E14.5 with placental iron accumulation, whereas inactivation in all embryonic tissues except the placenta is embryonic lethal at E17.5 with less placental iron accumulation. The data from the ZIP8 mice suggest that it’s not required for iron import but has an important yet unknown role in iron delivery to the fetus.
  12. Page 5 line 236-237: Endosome was used twice in this sentence.
  13. Page 6 first paragraph: What happens to iron once in the cytoplasm? Is it all exported via FPN or is some stored in ferritin? The authors should consider mentioning the candidate proteins for iron transport within the cytoplasm (i.e., PCBP proteins) and the candidate ferroxidases in the placenta (i.e., ceruloplasmin, hephaestin, zyklopen)
  14. Page 6 second paragraph: DMT1 is dispensable in the placenta so I am not sure it is correct to say that iron transport via DMT1 is enhanced. There could be other transporters like ZIP14 and ZIP8 that mediate DMT1-independent transport.
  15. Page 6 third paragraph: Sangkhae et al JCI should also be cited for lowering of hepcidin during pregnancy. The authors mention placental hepcidin may also regulate iron transport and thus the authors should provide evidence for or against that statement. Sangkhae et al (JCI and Blood) also demonstrate that fetal hepcidin does not regulate placental iron transporters in a global hepcidin KO model and should be cited in addition to the model in Kammerer et al.
  16. Page 6 line 280: Sangkhae et al Blood also show embryo hepcidin mRNA is lower than maternal.
  17. Page 6 lines 283-284: The authors should cite the research articles that demonstrate fetal hepcidin is increased by inflammation and infection (i.e., Fisher et al. JCI Insight. 2020;5(4):e135321; Tabbah et al. Am J Perinatol. 2018;35(9):865–872).
  18. Page 9 section 4: It needs to be clarified that Hmox1 encodes the HO-1 protein. Perhaps in the sentence on line 308 “… heme oxygenase 1 (HO1, encoded by Hmox1)…”
  19. Page 9 lines 312-313: This sentence has “well characterized” twice.
  20. Page 9 lines 317-321: The sentences describing how HO1 expression determines the attaching ability of the blastocyst, and HO1 expression in early embryos and placentas need citations.

Author Response

In this review, Kopec et al describe the current knowledge about the genes involved in iron homeostasis in mouse development. This review provides an interesting perspective of iron homeostatic genes in pre-implantation and post-implantation development. The authors rely largely on studies of transgenic (mainly knock-out) mouse models. The review is generally well written but contains several grammatical and spelling errors. Furthermore, the authors should include discussion on a broader selection of genes involved in iron homeostasis (see suggestions below) in order to provide a more comprehensive overview.

Our review has undergone English language editing by MDPI. Please find enclosed an appropriate certificate.

In general, as stated in the title, in our review we have included the information related to iron metabolism genes, deletion of which resulted in some perturbations of prenatal development  in mice. As regards other genes, we have no sufficient information to fully discuss their role in developing embryo/fetus.

  1. Page 1, lines 30-32: Citation #1 (Wilkinson and Pantopoulos Blood 2013) is about IRP and HIFs. The authors should consider adding a different reference, such as a review, which would be more appropriate.

Citation # 1 is not completely off the mark, however, we have added an additional reference.

  1. Page 2 lines 49-50: Ferroportin is expressed on many other cell types that contribute to systemic iron balance in addition to the duodenum and macrophages. The authors should include the release of iron from liver stores as another major source of iron export to plasma.

This has been done according to the Reviewer’s suggestion and an additional reference has been added (Zhang et al., 2012)

  1. Page 2 lines 59-61: What do the authors mean by “quantitatively differentiated iron acquisition?” This sentence should be clarified. Do the authors mean that the iron requirements depend on the developmental stage? The next sentence should also be clarified: “erythropoiesis” itself means the production of red blood cells so it is redundant to say erythropoiesis must produce more and more red blood cells.

Two sentences mentioned by the Reviewer have been clarified.

  1. Page 2 line 75: The authors should consider revising the statement that mice strongly replicate human iron metabolism. There are fundamental differences in iron metabolism between mice and humans. Humans rely primarily on iron recycling to meet daily requirements whereas mice depend mostly on dietary iron. However, the key regulatory and transport mechanisms are what is similar to humans.

The life span of the erythrocytes was found to be 40.7 ± 1.9 (S.D.) days in the mouse (app. 3 times shorter than in the man). This means that recycling of iron in the mouse is of non-negligible importance. We have based our statement on a very solid review by Nancy Andrews. Anyway, according to the Reviewer’s suggestion we have decided to attenuate this statement by deleting the word “strongly”.

  1. Page 3 first paragraph: The authors should use the official gene symbols for FTH1 (Fth1) and FTL (Ftl) and be consistent with the acronyms.

Done according to the Reviewer’s requirement.

  1. Page 3 lines 93-96: Is it known that the embryo meets its own iron demands in the preimplantation period? The iron maternal requirements during the first trimester are lower than in menstruating women (Bothwell et al Am J Clin Nutr 2000), so is it that the mother has sufficient iron stores during the preimplantation period to meet early embryo iron demands? Similarly, the authors need to provide citations for the statements that the oocyte contains high iron levels, and that development takes place in an iron-rich environment.

All issues raised by the Reviewer in this point have been addressed and some additional evidence attesting “iron autonomy”  of the embryo in the early preimplantation period has been added.

  1. Page 3 lines 137-141: Although the Hpx null mice die shortly after birth, they are born viable which suggests another iron source is utilized for development when transferrin is not available (unless there is residual transferrin still being produced that is enough to support development). The authors should at least mention that non-transferrin bound iron could be a likely iron source for the fetus in these conditions, however the mechanisms of NTBI transport are not known. NTBI is detectable in the fetus at least in the first trimester (Evans et al Mol Hum Reprod. 2011 Apr; 17(4):227-32).

The sentence about the possible role of NTBI as an iron source in Hpx null newborns has been       added.

  1. Page 4 lines 177 and 189: The authors state that the mature placenta appears by E14.5 in mice, however the mouse placenta is fully formed by E10.5.

According to J. Rossant, J. C. Cross, Placental development: Lessons from mouse mutants. Nat. Rev. Genet. 2, 538548 (2001). The mature placenta consists of three layers: the labyrinth, the spongiotrophoblast, and the maternal decidua. The fully matured placenta is assumed to be from E14.5. Hemberger, M., Hanna, C.W. & Dean, W. Mechanisms of early placental development in mouse and humans. Nat Rev Genet 21, 27–43 (2020).

  1. Page 5 line 203: It is not definitively known that DMT1 is the endosomal iron transporter in placenta, it could still be ZIP8 or ZIP14.

ZIP8 or ZIP14 have been added as potential apical iron transporters in the placenta.

  1. Page 5 line 216: Iron released from transferrin is the reduced before being exported out of the endosome. The candidate ferrireductases in the placenta should be mentioned (i.e., STEAP3, STEAP4)

We have added one sentence about STEAP3

  1. Page 5 last paragraph: In addition to DMT1, there are other potential transporters involved in iron trafficking (i.e., ZIP8, ZIP14) that are worth mentioning. Whereas ZIP14 appears to have a redundant role similar to DMT1, ZIP8 global knockout is embryonic lethal at E14.5 with placental iron accumulation, whereas inactivation in all embryonic tissues except the placenta is embryonic lethal at E17.5 with less placental iron accumulation. The data from the ZIP8 mice suggest that it’s not required for iron import but has an important yet unknown role in iron delivery to the fetus.

Potential role of ZIPs in iron transport has been described.

  1. Page 5 line 236-237: Endosome was used twice in this sentence.

This has been corrected.

  1. Page 6 first paragraph: What happens to iron once in the cytoplasm? Is it all exported via FPN or is some stored in ferritin? The authors should consider mentioning the candidate proteins for iron transport within the cytoplasm (i.e., PCBP proteins) and the candidate ferroxidases in the placenta (i.e., ceruloplasmin, hephaestin, zyklopen)

As stated above our review is not focused on all genes of iron metabolism. We have no information about iron disorders during prenatal mouse development being a consequence of the deletion of  many iron-related genes including those encoding PCBP proteins and copper-dependent ferroxidases and others. We believe that this review is addressed to readers with at least fundamental knowledge of molecular basis of iron metabolism and description of the role of all iron proteins is not necessary.

  1. Page 6 second paragraph: DMT1 is dispensable in the placenta so I am not sure it is correct to say that iron transport via DMT1 is enhanced. There could be other transporters like ZIP14 and ZIP8 that mediate DMT1-independent transport.

The role of ZIP proteins in iron transport across the placenta has been already addressed in response the Reviewer’s suggestion (point 11). Regarding DMT1, although it has been shown dispensable in the placenta, its strong expression in this tissue does not definitely excludes its role in iron transport under certain conditions (for ex. severe iron deficiency). 

  1. Page 6 third paragraph: Sangkhae et al JCI should also be cited for lowering of hepcidin during pregnancy. The authors mention placental hepcidin may also regulate iron transport and thus the authors should provide evidence for or against that statement. Sangkhae et al (JCI and Blood) also demonstrate that fetal hepcidin does not regulate placental iron transporters in a global hepcidin KO model and should be cited in addition to the model in Kammerer et al.

Sangkhae et al JCI and Blood have been cited according to the Reviewer’s requirement.

  1. Page 6 line 280: Sangkhae et al Blood also show embryo hepcidin mRNA is lower than maternal.

Sangkhae et al Blood  has been cited according to the Reviewer’s requirement.

  1. Page 6 lines 283-284: The authors should cite the research articles that demonstrate fetal hepcidin is increased by inflammation and infection (i.e., Fisher et al. JCI Insight. 2020;5(4):e135321; Tabbah et al. Am J Perinatol. 2018;35(9):865–872).

Done according to the reviewr’s suggestion.

  1. Page 9 section 4: It needs to be clarified that Hmox1 encodes the HO-1 protein. Perhaps in the sentence on line 308 “… heme oxygenase 1 (HO1, encoded by Hmox1)…”

This has been corrected according to the Reviewer’s suggestion

  1. Page 9 lines 312-313: This sentence has “well characterized” twice.

This has been corrected

  1. Page 9 lines 317-321: The sentences describing how HO1 expression determines the attaching ability of the blastocyst, and HO1 expression in early embryos and placentas need citations.

One reference has been added.